# DFT–Assisted Structure Determination from Powder X-ray Diffraction Data of a New Zonisamide/ε-Caprolactam Cocrystal

Rafael Barbas [1], Anna Portell [1], Rafel Prohens [1,*] and Antonio Frontera [2,*]

1   Unitat de Polimorfisme i Calorimetria, Centres Científics i Tecnològics, Universitat de Barcelona,
    Baldiri Reixac 10, 08028 Barcelona, Spain; rafa@ccit.ub.edu (R.B.); anna.portell@gmail.com (A.P.)
2   Departament de Química, Universitat de les Illes Balears, Crta. de Valldemossa, km 7.5, 07122 Palma, Spain
*   Correspondence: rafel@ccit.ub.edu (R.P.); toni.frontera@uib.es (A.F.)

**Abstract:** The crystal structure of a new zonisamide cocrystal, an anticonvulsant drug used to treat the symptoms of epilepsy and Parkinson's disease, with ε-caprolactam is reported herein. The structure has been solved by direct space methodologies from powder X-ray diffraction data. The refinement of the structure was conducted by the Rietveld method assisted by the dispersion-corrected density-functional theory (D-DFT) calculations and periodic boundary conditions. Further analysis of the structure reveals several H-bonded synthons and self–assembled dimers that have been further analyzed by DFT calculations and other computational tools such as molecular electrostatic potential (MEP) surfaces and the quantum theory of "atom-in-molecules" (QTAIM).

**Keywords:** powder X-ray diffraction; DFT calculations; zonisamide; hydrogen bonding; crystal engineering

## 1. Introduction

A convenient strategy to improve the physicochemical properties (solubility, stability, dissolution rate, etc.) of active pharmaceutical ingredients (APIs) is the utilization of cocrystals, as described in the literature [1]. Cocrystals are multicomponent molecular crystals that exhibit a potential advantage from the pharmaceutical point of view. That is, a significant change in the physicochemical properties of the API can be accomplished without its covalent modification. This technique has been used to improve pharmaceutically important properties of pharmaceutical and nutraceutical compounds in terms of hygroscopicity, bioavailability [2], stability [3,4], mechanical properties, etc. [5–7].

Zonisamide (1,2-benzoxazol-3-ylmethanesulfonamide, Figure 1) is an API that functions as an anticonvulsant [8]. Compared to other antiepileptic drugs, it has been demonstrated that it has lower hypnotic effect and toxicity [9]. However, the half-life of zonisamide is significantly reduced if administered with other drugs [10,11]. Cocrystallization has been proposed as a potential solution [12] and, consequently, several cocrystals of zonisamide have been synthesized using several coformers such as L-proline, catechol, 5-methyl-2-pyridone and caffeine, though only two cocrystals have resolved by single X-ray structure determination so far, which are zonisamide-caffeine and zonisamide-methylpyridone [12].



**Figure 1.** Zonisamide and ε-caprolactam, components of the title cocrystal.

This work reports the crystal structure of a new zonisamide/$\epsilon$-caprolactam cocrystal, (Figure 1). The structure was resolved using powder X-ray diffraction data and required the application of density-functional theory (DFT) calculations in order to achieve a reasonable structure. The final structure solution was determined by direct space methodologies starting from a molecular model optimized by DFT and several trials of twenty million runs as described in the following sections. The best solution was refined by the Rietveld method and, subsequently, the structure was subjected to optimization by the dispersion-corrected density-functional theory (D-DFT) calculations and periodic boundary conditions. The solid-state architecture of the cocrystal has been also analyzed by DFT calculations and several computational tools including molecular electrostatic potential (MEP) surfaces and the quantum theory of "atom-in-molecules" (QTAIM) focusing on the H-bonding interactions.

## 2. Materials, Experimental and Theoretical Methods

### 2.1. Materials

Anhydrous Zonisamide (provided by Urquima, S. A. Sant Fost de Campsentelles, Spain) (corresponding to the form with CCDC refcode VUXPUZ) and anhydrous $\epsilon$-caprolactam (from Sigma-Aldrich, St. Louis, MO, USA) (corresponding to polymorph I with CCDC refcode CAPLAC16) were used as starting material, (see Supplementary Materials for further details).

### 2.2. Sample Preparation

Synthesis of the bulk powder of zonisamide/$\epsilon$-caprolactam cocrystal was conducted by a liquid assisted grinding experiment (LAG) in tetrahydrofuran (THF). Details of the synthesis are as follows: zonisamide (20.0 mg, 0.094 mmol) and $\epsilon$-caprolactam (10.6 mg, 0.094 mmol) were ground together with one drop of THF using a Retsch MM 2000 grinding mill. The mixture was placed in 2 mL volume stainless steel jars, along with two stainless tungsten grinding balls of 3 mm diameter. Grinding was performed for 30 min, with a frequency of the mill of 30 Hz. Finally, the sample was collected immediately without prior drying for XRPD analysis. The formation of a new cocrystal was determined by comparing XRPD patterns of starting materials and products from the cocrystal screening LAG experiment (see Supplementary Materials for further details).

### 2.3. X-ray Powder Diffraction: Data Collection

Powder X-ray diffraction pattern of the zonisamide/$\epsilon$-caprolactam cocrystal was obtained on a PANalytical X'Pert PRO MPD diffractometer in transmission configuration using Cu $K\alpha_{1+2}$ radiation ($\lambda$ = 1.5418 Å) with a focusing elliptic mirror and a PIXcel detector working at a maximum detector's active length of 3.347°. We configured a convergent beam with a focalizing mirror and a transmission geometry with flat sample sandwiched between low absorbing films measuring from 2 to 70° in 2$\theta$, with a step size of 0.013° 2$\theta$ and a total measuring time of 47.5 h at room temperature (298 K). Atomic coordinates for the zonisamide/$\epsilon$-caprolactam cocrystal have been deposited with the Cambridge Crystallographic Data Centre (CCDC number is 2189669). The supplementary crystallographic data can be obtained free of charge from the Cambridge Crystallographic Data Centre via www.ccdc.cam.ac.uk/data_request/cif (accessed on 12 July 2022).

### 2.4. DFT Calculations

The best solution from the direct space methodology conducted with the FOX software of zonisamide/$\epsilon$-caprolactam cocrystal was subjected to geometry optimization by DFT [13,14] periodic calculations performed within the generalized gradient approximation (GGA) [15], as provided by the module CASTEP [16] in Materials Studio software [17], using a basis set cutoff energy of 520 eV, ultrasoft pseudopotentials [18], PBE functional [15], semi-empirical dispersion corrections (Grimme) [19] and fixed unit cell, a 1 × 6 × 2 k-point Monkhorst-Pack grid [20] was used with experimental *C*2/*c* space group and periodic boundary conditions. The validation of the structure [21,22] was conducted through a

second calculation starting from the optimized structure obtained in the previous step by using the same DFT-D parameters but setting free the unit cell parameters. Atomic displacement RMSD values were calculated to assess the reliability of the optimization.

The DFT study of the supramolecular assemblies observed in the solid state was performed using the Gaussian-16 [23] program and the PBE0-D3/def2-TZVP level of theory [24,25]. The interaction energies were estimated by calculating the difference between the absolute energies of isolated monomers and their assembly. The BSSE (basis set superposition error) correction was applied to the interaction energies reported in this manuscript using the method proposed by Boys–Bernardi [26]. The Bader's "Atoms in molecules" theory (QTAIM) [27] was used to further characterize the noncovalent contacts observed in the cocrystal using the AIMAll calculation package [28]. The MEP surface plots were generated using the Gaussian-16 software [23] and the 0.001 a.u. isovalue for the density as a best estimate of the van der Waals envelop.

## 3. Results and Discussion

### 3.1. Structure Determination from Space Direct Methods

#### 3.1.1. Structure Determination and Initial Rietveld Refinement

The structure determination of zonisamide/$\epsilon$-caprolactam cocrystal from powder X-ray diffraction data was carried out using a high quality, good resolution and high statistics, transmission powder diffraction data. Its powder XRD pattern was indexed and the lattice parameters were refined by means of Le Bail fits using DICVOL04 [29,30] with the following monoclinic cell values: a = 32.150(4) Å, b = 5.3272(5) Å, c = 18.730(2) Å, $\beta$ = 95.623(6)°, V = 3192.5(6) Å$^3$, and a number of impurities equal to zero. *C2/c* space group was deduced from the systematic absences and confirmed with the SGAid program of the DAJUST [31] software. Le Bail fit ($R_{wp}$ = 4.18%, $R_p$ = 2.46%) (see Supplementary Materials for further details). According to the estimated density (1.35 Mg m$^{-3}$), the asymmetric unit was assumed to contain one molecule of zonisamide and one molecule of $\epsilon$-caprolactam (Z = 8). The structure solution was carried out by direct space methodologies starting from a molecular model optimized by DFT with SPARTAN [32] by means of the program FOX [33] with the parallel tempering algorithm. Some constraints were introduced to FOX, considering benzisoxazole ring as rigid group. Several trials of 20 million runs were performed. The best solution (based on the $R_{wp}$ value) was refined by the Rietveld method using FullProf [34], Figure 2 depicts the final Rietveld plot of this first structure. Subsequently, the structure was subjected to optimization by the dispersion-corrected density-functional theory calculations.

#### 3.1.2. Crystal Structure Optimization

Since the Rietveld plot shows relevant differences between the experimental and calculated XRPD diagram in the 8–22° 2$\theta$ range (mainly the diffraction peak at 9.5° in 2$\theta$) the bonds distances and angles of the structural model of zonisamide/$\epsilon$-caprolactam cocrystal were analyzed with Mogul 1.8.5 (The Cambridge Crystallographic Data Centre, Cambridge, UK) [35,36] in order to check if the bond lengths and angles were consistent with the CCDC database. The best solution shows some bond distances and angles associated to the benzisoxazole ring which are unusual (see Supplementary Materials, Tables S1 and S2 for further details). Moreover, the checkcif shows an alert C level about the absence of acceptor for N24-H35 bond (see Supplementary Materials, Figure S15 for further details). Thus, in order to improve these structural arrangements, DFT-D geometry optimization was carried out in three steps: in the first optimization, all coordinate positions were fixed except O23, N25 and C27; in the second optimization, all coordinate positions were fixed except C26 and finally in the third optimization, only S20, O21 and O22 were computed, in all cases the cell parameters were kept fixed. The final Rietveld plot of this optimization shows a better agreement ($R_{wp}$ = 8.20%, $R_p$ = 5.17%) compared to the first one (see Supplementary Materials, Figure S6). Thus, this optimized structure was used as the initial structural model

for a DFT-D optimization of all of the atomic positions with fixed unit cell parameters using CASTEP.

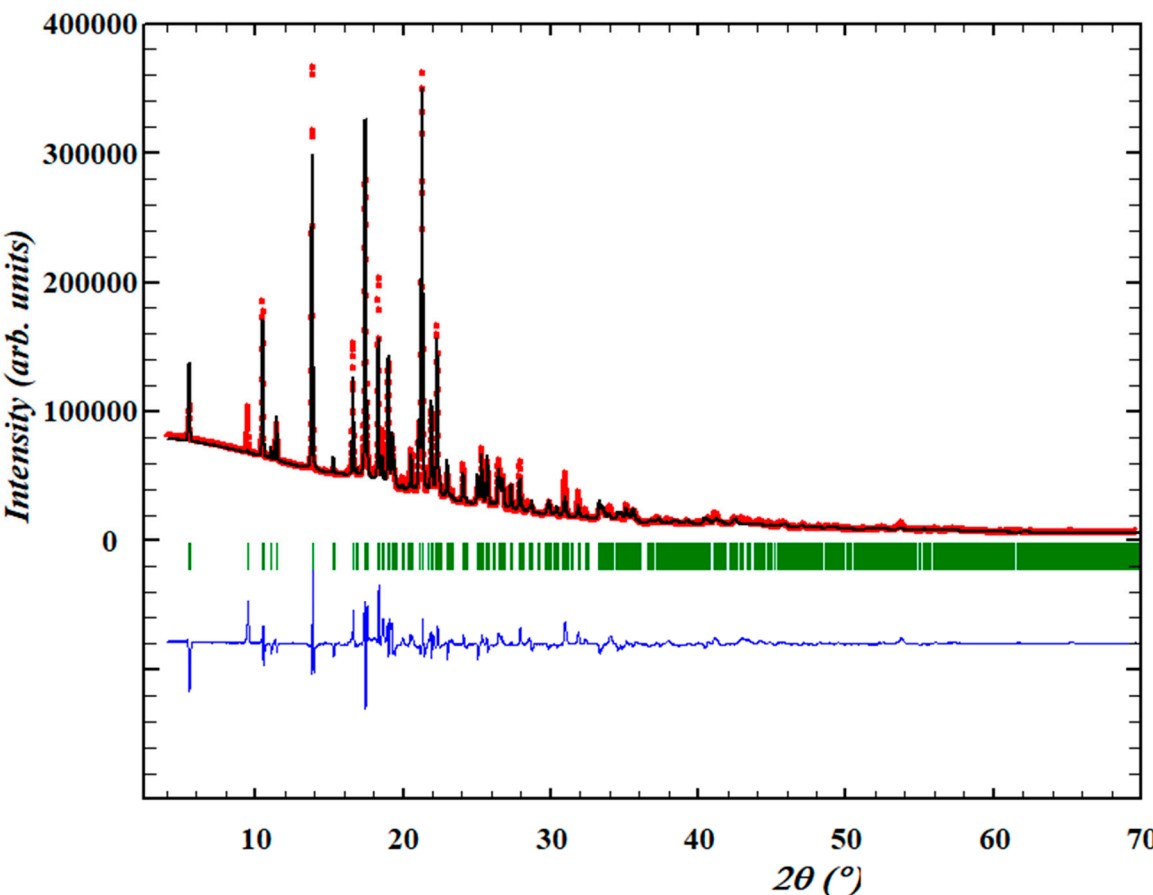

**Figure 2.** Rietveld plot for the best solution of zonisamide/ε-caprolactam cocrystal obtained by FOX. The plot shows the experimental powder XRD profile (red marks), the calculated powder XRD profile (black solid line), the difference profile (blue, lower line) and Bragg positions (green lines). Agreement factors: $R_{wp}$ = 8.32%, $R_p$ = 5.30%. Enlargement from 8 to 22° in $2\theta$ is also represented.

The crystal structure obtained after 30 optimization steps shows a worst agreement ($R_{wp}$ = 12.7%, $R_p$ = 7.15%) (see Supplementary Materials, Figure S11) compared to the starting point. Particularly relevant is the big difference at the 9.5°, 11.1°, 13.8° and 15.3° in $2\theta$ diffraction peaks, which correspond to the (002), (400), (40-2) and (402) hkl planes, respectively (Figure 3).

An analysis of the atoms involved in such diffraction planes reveals that the H-bonded region between the zonisamide and ε-caprolactam molecules is the most affected by the DFT calculation. In particular, the (002) plane at 9.5° in $2\theta$ contains the atoms involved in both ε-caprolactam dimer and isoxazole moiety of the zonisamide dimer and it was chosen as a diagnostic peak. Thus, in order to select among all of the optimization steps the one which should conduct the final Rietveld refinement, a combination of criteria was applied to structures from steps 9, 10, 11 and 20: (a) $R_{wp}$ Rietveld refinement, (b) good agreement at the 9.5° diffraction peak intensity and (c) good angles and bonds values according to Mogul analysis. Figure 4 shows the DFT optimization energy profile and Figure 5 shows the calculated powder XRD profile from Rietveld refinements for the starting point and the steps 11 and 30 compared with the experimental powder pattern of the cocrystal. Tables S1 and S2 of the Supplementary Materials section contain the Mogul Geometry check. In addition, the self-assembling of ε-caprolactam by hydrogen bonding was analyzed in the CCDC, Figures S13 and S14 Supplementary Materials shows N-H⋯O

distances of the ε-caprolactam dimer of the different optimization steps compared to the mean value from the CCDC analysis. Taking into account all of these criteria, the structure from step 11 was chosen for subsequent Rietveld refinement. Figure 6 shows the final Rietveld plot.

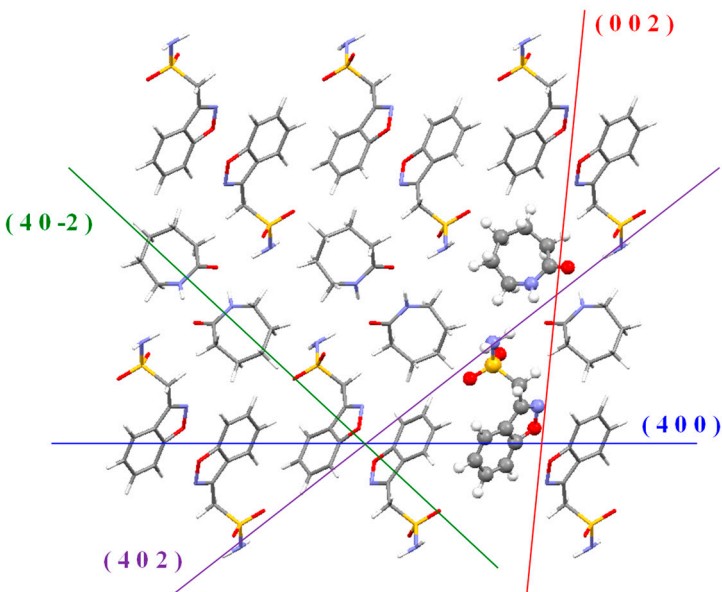

**Figure 3.** Relevant lattice planes (hkl). Asymmetric unit is shown in ball-and-sticks representation.

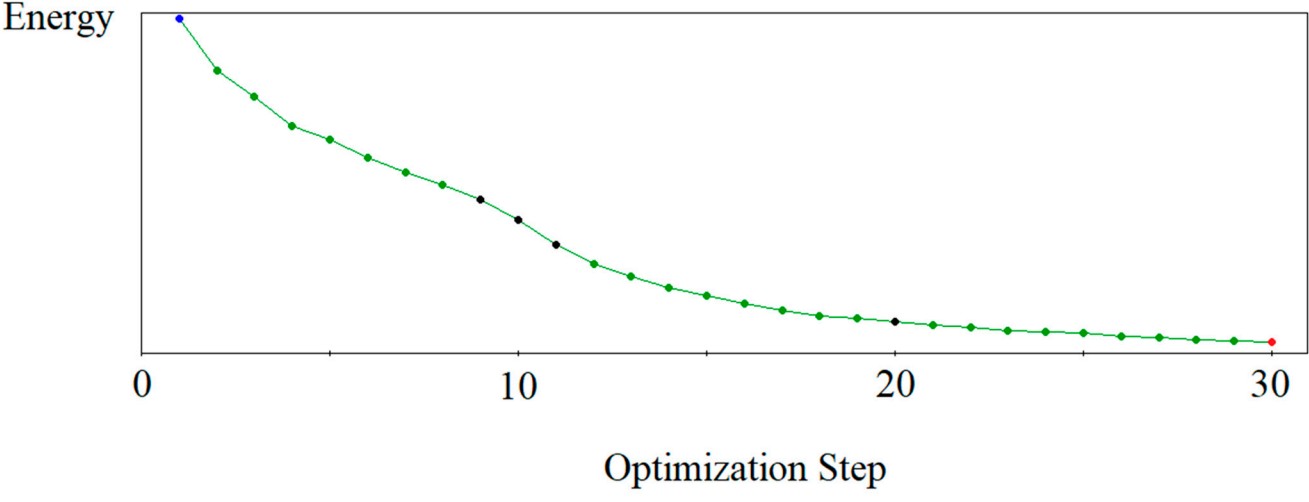

**Figure 4.** DFT optimization energy profile. Relevant optimization steps are colored: blue for the initial optimization step, black for 9, 10, 11 and 20 optimization steps, respectively and red for the final optimization step.

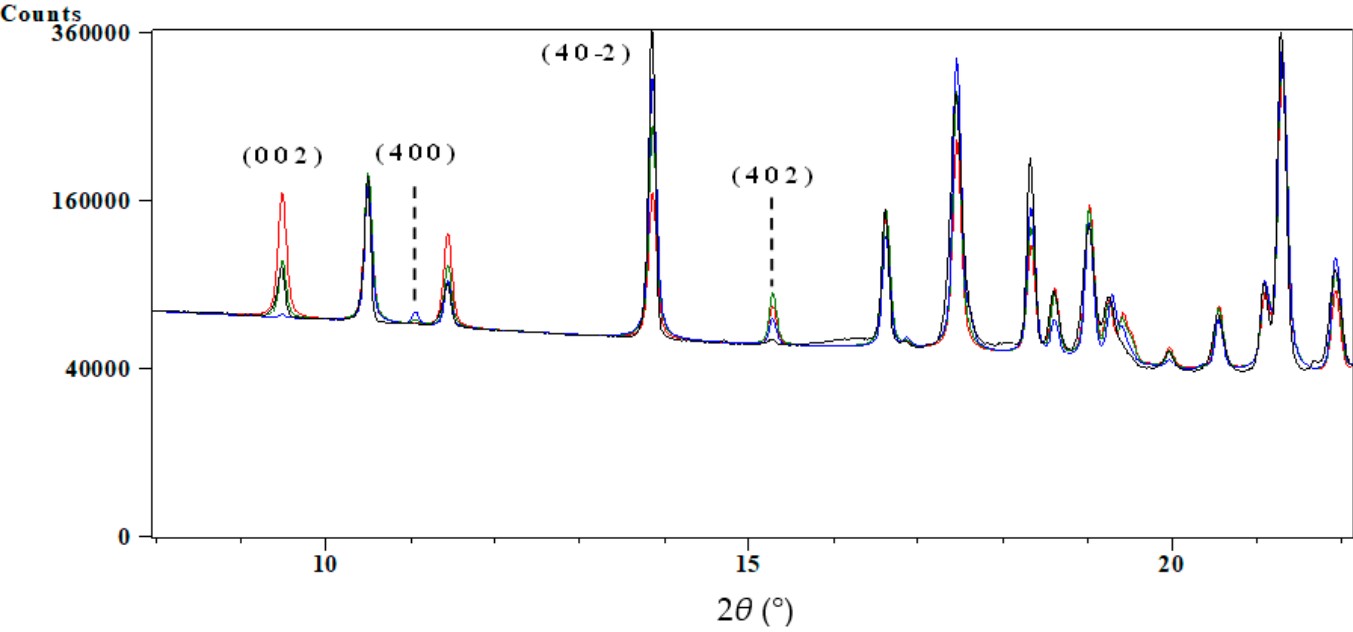

**Figure 5.** Comparison between XRPD diffractograms of the experimental bulk powder (black) and the calculated powder XRD profiles from Rietveld refinements (structure models before optimization (blue), after DFT-D geometry optimization: step 11 (green) and step 30 (red)). Only enlargement from 8 to 22° 2θ is represented. Most relevant diffraction peaks (hkl) are identified.

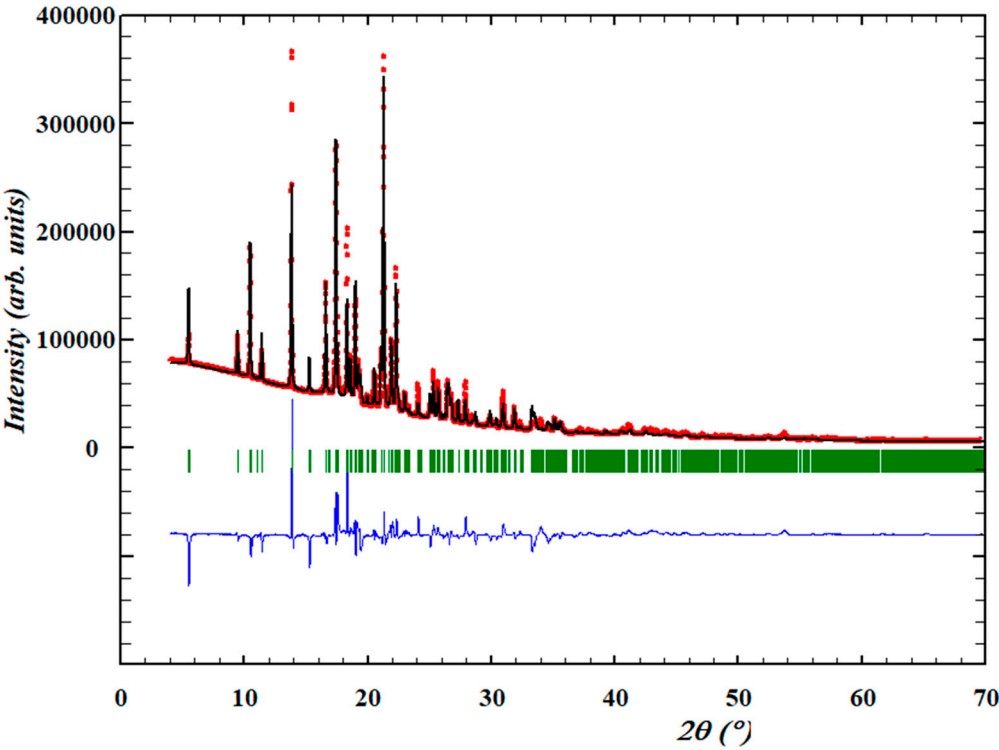

**Figure 6.** Rietveld plot for the structure from step 11 of zonisamide/ε-caprolactam cocrystal obtained by DFT-D. The plot shows the experimental powder XRD profile (red marks), the calculated powder XRD profile (black solid line), the difference profile (blue, lower line) and Bragg positions (green lines). Agreement factors: $R_{wp}$ = 9.66%, $R_p$ = 5.62%. Enlargement from 8 to 22° in 2θ is also represented.

### 3.1.3. Crystal Structure Validation by DFT-D

Finally, the RMS Cartesian displacement of the non-hydrogen atoms were calculated, being 0.042 Å and 0.035 Å for zonisamide and ε-caprolactam molecules, respectively, showing excellent agreement between the structure refined from step 11 (which is the finally reported one) and the DFT-D optimized free cell structure, Figure 7; this thus provided a good validation check for this crystal structure (see Supplementary Materials for further details) [21].

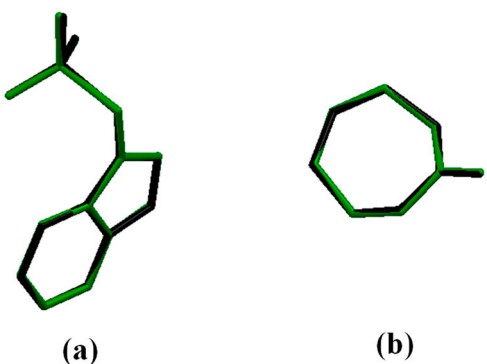

**(a)**               **(b)**

**Figure 7.** Overlay of the non-hydrogen atoms for zonisamide (**a**) and ε-caprolactam (**b**) molecules between the structure from the step 11 (green) and the DFT-D optimized free cell parameters structure (black).

### 3.2. DFT Calculations of the Supramolecular Assemblies

The DFT study of supramolecular assemblies conducted in this work was essentially focused on the analysis of the H-bonded network shown in Figure 8a, where the ε-caprolactam units self-assembled via two symmetrically equivalent N–H⋯O. The resulting centrosymmetric dimer interacts with two zonisamide molecules by the formation of additional NH⋯O and CH⋯O H-bonds. In addition, Figure 8b shows the formation of 1D infinite chains propagated by NH⋯O H-bonds involving the sulfonamide groups. These 1D-chains are interconnected by CH⋯O H-bonds [$R_2^2(8)$ synthon] and π-stacking interactions.

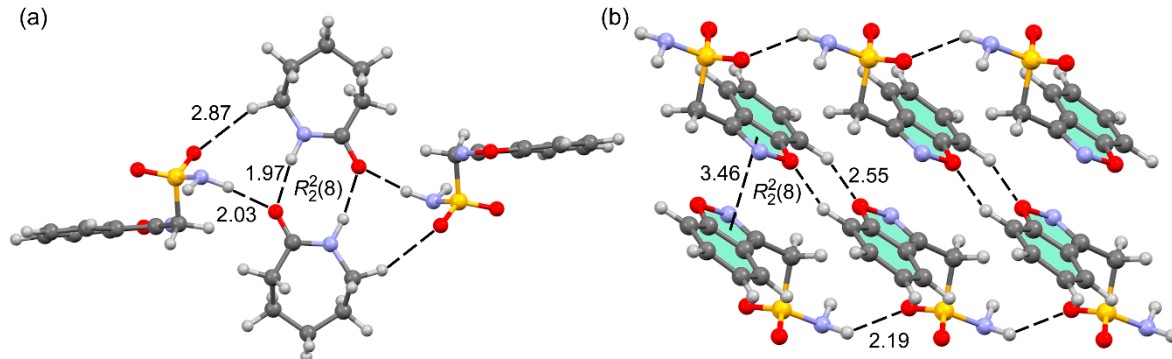

**Figure 8.** (**a**) Partial view of the cocrystal showing the H-bonding dimer of ε-caprolactam and two appended zonisamide molecules. (**b**) 1D supramolecular chains of H-bonded zonisamide molecules connected by $R_2^2(8)$ synthons and π-stacking interactions. Distances in Å.

Firstly, the MEP surfaces of both compounds were computed (see Figure 9) to investigate the MEP values at the H-bond donor and acceptor groups and rationalize the formation of the H-bonds shown in Figure 8. For zonisamide (Figure 9a), the MEP minima are located at the O-atoms of the sulfonamide group (−38.3 kcal/mol) followed by the N-atom of the isoxazole ring (−31.4 kcal/mol). The MEP values are also negative at the O-atom of the isoxazole ring (−21.3 kcal/mol) and over the six-membered ring (−13.2 kcal/mol). The

maximum MEP is found at the NH$_2$ group (+55.8 kcal/mol) of the sulfonamide group. Furthermore, the MEP at the H-atoms in alpha to the sulfonamide group are also large and positive (38.4 kcal/mol) in line with their expected acidity. For ε-caprolactame (Figure 9b), the MEP minima is located at the O-atom of the amide group (−48.3 kcal/mol) and the maximum MEP is found at the amidic NH (+34.5 kcal/mol). Taken together, the MEP maximum is found in the zonisamide molecule and the minimum at the ε-caprolactam, thus the most favored interaction between both coformers corresponds to NH···O(amide). This MEP analysis is useful to explain the formation of the H-bonding network represented in Figure 8a and the 1D chain in Figure 8b, since the strongest H-bond donor (sulfonamide NH$_2$) interacts with the most nucleophilic O-atom (ε-caprolactam) and the second most nucleophilic O-atom (sulfonamide group) via both H-atoms.

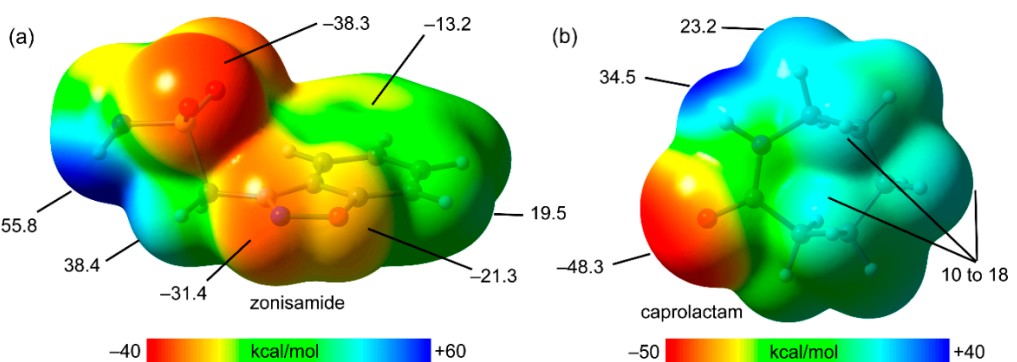

**Figure 9.** MEP surfaces of zonisamide (**a**) and ε-caprolactam (**b**) at the PBE0-D3/def2-TZVP level of theory (density isovalue 0.001 a.u.). The energies are given in kcal/mol.

The QTAIM distribution of the H-bonded tetrameric assembly represented in Figure 10a evidences that each H-bond is characterized by a bond critical point (CP, green sphere) and the bond path interconnecting the H and O-atoms (see Table S3 for the QTAIM parameters at the bond CPs). This analysis also confirms the existence of the ancillary CH···O contact involving one C(sp$^3$)–H bond of the ε-caprolactam. In fact, the MEP analysis disclosed that the H-atoms of the methylene group in alpha with respect to the amino group are more positive (23.2 kcal/mol, see Figure 9b) than the rest of methylene groups. The total binding energy of this tetramer is very large, 50.2 kcal/mol, thus confirming the importance of these H-bonds for the formation of the cocrystal. The QTAIM of the homotetrameric assembly is provided in Figure 10b, where only zonisamide molecules are involved. In this case, the binding energy, 36.4 kcal/mol, is smaller in absolute value due to the presence of only two strong N–H···O H-bonds. The QTAIM analysis reveals the existence of π-stacking interactions characterized by four bond CP and bond paths interconnecting three atoms of the five membered ring and one atom of the six membered ring (see Table S3 for selected QTAIM parameters at the bond CPs). It can be observed that the zonisamide rings are connected by two CH···O H-bonds and one O···O interaction (marked with a yellow circle) that is expected to be repulsive taking into consideration the negative MEP value at the O-atom (see Figure 9a), although part of the negative charge is delocalized due to the formation of the CH···O contacts. This O···O interaction is confirmed by the presence of one bond CP and bond path interconnecting both O-atoms. At this point, it should be mentioned that the existence of a bond CP and a bond path is clear evidence of an interaction, which can be attractive or repulsive. In fact, bond CPs have been found in many counterintuitive cases, which most often correspond to the presence of bond CPs between two electronegative atoms [37,38].

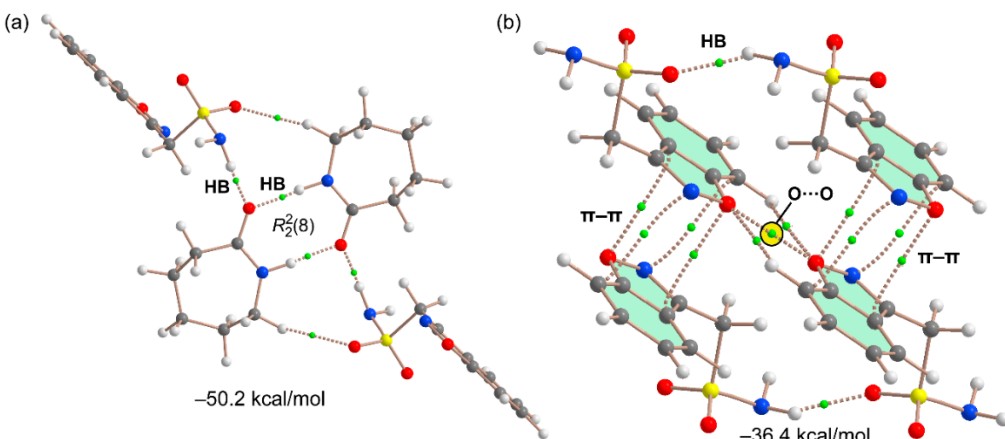

**Figure 10.** QTAIM analysis of intermolecular bond CPs (green spheres) and bond paths for the hetero (**a**) and homotetrameric (**b**) assemblies observed in the zonisamide/ε-caprolactam cocrystal.

## 4. Concluding Remarks

In summary, our work reports the crystal structure of a new zonisamide cocrystal solved by means of direct space methodologies from powder X-ray diffraction data and assisted by DFT calculations, enriching the scarce list of cocrystals of this important drug compound in the literature. Our computational analysis by DFT reveals that the H-bonds between both API and coformer are very strong and the MEP surface study corroborates the presence of strong H-bond donor and acceptor groups in both compounds. In addition, zonisamide molecules form 1D supramolecular chains in the solid state propagated by NH···O H-bonds that are interconnected by π-stacking and CH···O H-bonds. Moreover, most likely repulsive O···O contacts also exist between zonisamide molecules, as demonstrated by QTAIM analysis; but this is compensated by other attractive interactions in the same region such as CH···O H-bonds, which allows for a justification of the short O···O contact alerted by the check CIF validation procedure. Crystal structure solution from direct space methods is already a mature field but our work illustrates, with a new example, the value of DFT calculations to help solve crystal structures from XRPD data in those cases where the process is not straightforward.

**Supplementary Materials:** The following are available online at https://www.mdpi.com/article/10.3390/cryst12081020/s1, Figure S1: Comparative XRPD diffractograms between bulk powder of zonisamide and simulated from the cif file. Figure S2: Comparative XRPD diffractograms of cocrystal bulk powder, zonisamide bulk powder and ε-caprolactam bulk powder polymorph I. Figure S3: Le Bail fit from zonisamide/ε-caprolactam cocrystal. Figure S4: Label atom representation corresponding to the asymmetric unit of zonisamide/ε-caprolactam cocrystal. Tables S1 and S2: The bond distances and the angles of the structural models of zonisamide/ε-caprolactam cocrystal, respectively. Table S3. Density ($\rho$), $\lambda 1$, $\lambda 2$, $\lambda 3$ and Laplacian $\nabla 2\rho$ of electron density at the bond CPs of the noncovalent contacts of the cocrystal (see Figure 10, main text) in atomic units. Figures S5–S11: Rietveld plots for the best solution obtained by FOX, the initial structure model after some structural arrangements, the structural model computed in the optimization step number 9, the structural model computed in the optimization step number 10, the structural model computed in the optimization step number 11, the structural model computed in the optimization step number 20 and the structural model computed in the optimization step number 30, respectively. Figure S12: Comparison between XRPD diffractograms of bulk powder zonisamide/ε-caprolactam cocrystal and the calculated powder XRD profile from Rietveld refinements (structure models before and after DFT-D geometry optimization: optimization step 9, optimization step 10, optimization step 11, optimization step 20 and optimization step 30 structure model. Figure S13: N-H···O distances of the self-assembling of ε-caprolactam dimer reported in the CCDC. Figure S14: Overlay of the crystal structures of the different structures before and after optimization steps. N-H···O distances measured are represented. Figure S15: C-H···O23 and O23···O23 distances measured of the crystal structures of the different structures before and after optimization steps are represented. More relevant

Alert level reported from each checkcif crystal structures are also reported. Figure S16: The RMS Cartesian displacement of the non-hydrogen atoms for zonisamide and ϵ-caprolactam molecules in the asymmetric unit between the DFT-optimized free cell parameters (black) and different structures.

**Author Contributions:** Conceptualization, A.F. and R.P.; methodology, R.B., R.P., A.P. and A.F.; validation, R.P., R.B.; formal analysis, R.B., R.P. and A.F.; investigation, A.F., R.P. and R.B.; resources, R.P. and A.F.; data curation, R.B.; writing—original draft preparation, A.F. and R.P.; writing—review and editing, A.F., R.B. and R.P.; supervision, A.F. and R.P.; project administration, A.F. and R.P.; funding acquisition, A.F. and R.P. All authors have read and agreed to the published version of the manuscript.

**Funding:** This research was funded by MICIU/AEI of Spain (project PID2020-115637GB-I00 FEDER funds).

**Acknowledgments:** We thank the "centre de tecnologies de la informació" (CTI) at the University of the Balearic Islands for computational facilities.

**Conflicts of Interest:** The authors declare no conflict of interest.

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
