# Peer review of "DFT–Assisted Structure Determination from Powder X-ray Diffraction Data of a New Zonisamide/ϵ-Caprolactam Cocrystal"

_crystals, doi:10.3390/cryst12081020_

Round 1
Reviewer 1 Report
The manuscript concerns the synthesis and structure description of the new zonisamide/ϵ-caprolactam cocrystal. This topic is exciting for the pharmaceutical application of zonisamide as an anticonvulsant because the half-life of this matter is significantly reduced if administered with other drugs, and its cocrystallization with some other substances is proposed as a solution to this problem. The authors described the method of zonisamide/ϵ-caprolactam cocrystal synthesis and the results of the Powder X-Ray Diffraction and DFT studies of its structure. These results seem reliable, and I believe that manuscript must be published. However, it needs the following minor revisions.
1) Authors demonstrate, with the help of QTAIM analysis, the presence of bond critical points describing the formation of the hydrogen bonds, π-π-interaction, and O×××O interaction in the hetero and homotetrameric assemblies observed in the zonisamide/ϵ-caprolactam cocrystal (Figure 10). Usually, such analysis is accompanied by representing the topological parameters of the electron density in these points, such as the values of the electron density and its Laplacian as well as Laplacian’s eigenvalues λ1, λ2, and λ3. I recommend adding a table with these values into the main text or ESI (maybe ESI is even better). It will allow other researchers to compare and interpret their results dealing with the same types of interactions.
2) Describing the type of O×××O interaction (attraction or repulsion) in zonisamide dimer, the authors use equation E = –233.1 × ρ + 0.7 to estimate the hydrogen bond energy, assuming that the total binding energy of two monomers consists of the binding energies of two CH×××O H-bonds and energy of O×××O interaction. In the paper [37], authors cited, this equation looks as E = –223.1 × ρ + 0.7 (or E = –223.08 × ρ + 0.7423 in original). There is a difference in the constant before electron density (233.1 vs. 223.1).
Is it a typo? Or did the authors use the value 233.1 for calculations? It can be a problem because the mean absolute percentage error of the used equation, estimated by its authors Emamian et al. [37], is 14.7%. It means that the energy of O×××O interaction must be calculated in some diapason, and it is unclear whether both limits of this diapason are positive.
Moreover, Emamian et al. have received mentioned equation from the calculations performed with the help of the B3LYP-D3/ma-TZVP method. In contrast, the authors used the PBE0-D3/def2-TZVP method here, i.e., another GGA functional and the same basis set, but without diffuse functions. Another method leads to another electron density distribution, resulting in another electron density value in the bond critical points. Therefore the mean absolute percentage error of the used equation can be higher than 14.7%. I recommend authors check and comment on this matter.
Author Response
We thank this referee for his/her careful reading of the manuscript important corrections and suggestions. The changes made are listed below and marked in the manuscript using a yellow background.
1) Authors demonstrate, with the help of QTAIM analysis, the presence of bond critical points describing the formation of the hydrogen bonds, π-π-interaction, and O×××O interaction in the hetero and homotetrameric assemblies observed in the zonisamide/ϵ-caprolactam cocrystal (Figure 10). Usually, such analysis is accompanied by representing the topological parameters of the electron density in these points, such as the values of the electron density and its Laplacian as well as Laplacian’s eigenvalues λ1, λ2, and λ3. I recommend adding a table with these values into the main text or ESI (maybe ESI is even better). It will allow other researchers to compare and interpret their results dealing with the same types of interactions.
Reply: Thank you for this suggestion, done!, see Table S3 in ESI
2) Describing the type of O×××O interaction (attraction or repulsion) in zonisamide dimer, the authors use equation E = –233.1 × ρ + 0.7 to estimate the hydrogen bond energy, assuming that the total binding energy of two monomers consists of the binding energies of two CH×××O H-bonds and energy of O×××O interaction. In the paper [37], authors cited, this equation looks as E = –223.1 × ρ + 0.7 (or E = –223.08 × ρ + 0.7423 in original). There is a difference in the constant before electron density (233.1 vs. 223.1).
Is it a typo? Or did the authors use the value 233.1 for calculations? It can be a problem because the mean absolute percentage error of the used equation, estimated by its authors Emamian et al. [37], is 14.7%. It means that the energy of O×××O interaction must be calculated in some diapason, and it is unclear whether both limits of this diapason are positive.
Moreover, Emamian et al. have received mentioned equation from the calculations performed with the help of the B3LYP-D3/ma-TZVP method. In contrast, the authors used the PBE0-D3/def2-TZVP method here, i.e., another GGA functional and the same basis set, but without diffuse functions. Another method leads to another electron density distribution, resulting in another electron density value in the bond critical points. Therefore, the mean absolute percentage error of the used equation can be higher than 14.7%. I recommend authors check and comment on this matter.
Reply: We are very grateful to this referee for taking all this to our attention, including the typo in the formula, the basis set and DFT method dependence. We were not aware of the error percentage of this method. We have decided to eliminate this part of the discussion from the manuscript since the main aim of this work is developing a method to resolve X-ray structures from XRPD and it is not to the study of this contact (a very secondary aspect of the work).
Reviewer 2 Report
see attached file

Reviewer 3 Report
By the example of a new zonisamide/ϵ-caprolactam cocrystal the advantages of combined DFT-assisted solution and refinements technique for powder X-ray diffractometry is demonstrated. The work is performed at high experimental and theoretical level. This research represents sufficient contribution to development of new approaches in crystal structure determinations from powder diffraction data.
Author Response
We thank this referee for his/her careful reading of the manuscript and encouraging comments